# Organic and Conventional Coffee Beans, Infusions, and Grounds as a Rich Sources of Phenolic Compounds in Coffees from Different Origins

**DOI:** 10.3390/molecules30061290

**Published:** 2025-03-13

**Authors:** Alicja Ponder, Karol Krakówko, Marcin Kruk, Sebastian Kuliński, Rafał Magoń, Daniel Ziółkowski, Elvyra Jariene, Ewelina Hallmann

**Affiliations:** 1Department of Functional and Organic Food, Institute of Human Nutrition Sciences, Warsaw University of Life Sciences, Nowoursynowska 159c, 02-776 Warsaw, Poland; alicja_ponder1@sggw.edu.pl (A.P.); karol_krakowko@sggw.edu.pl (K.K.); 2Department of Food Gastronomy and Food Hygiene, Institute of Human Nutrition Sciences, University of Life Sciences, Nowoursynowska 159c, 02-776 Warsaw, Poland; marcin_kruk@sggw.edu.pl; 3Faculty of Pedagogy and Health Education and Dietetics, The University of the West Indies, Cave Hill Rd., Box 1341, Wanstead BB11000, Barbados; sebastian.k.socrates@outlook.com; 4Department of Security Science, Faculty of Applied Sciences, Academy of the Higher School of Banking, Cieplaka 1C, 41-300 Dąbrowa Górnicza, Poland; r.magon@ron.mil.pl; 5Faculty of Electronics, gen. Sylwestra Kaliskiego 2, Military University of Technology, 00-908 Warsaw, Poland; daniel.z.socrates@outlook.com; 6Department of Plant Biology and Food Sciences, Agriculture Academy, Vytautas Magnus University, Donelaicio St. 58, 44248 Kaunas, Lithuania; elvyra.jariene@vdu.lt; 7Bioeconomy Research Institute, Agriculture Academy, Vytautas Magnus University, Donelaicio 58, 44248 Kaunas, Lithuania

**Keywords:** coffee beans, coffee brews, coffee grounds, conventional coffee, organic coffee, polyphenols

## Abstract

Coffee is a beverage that contains a high concentration of bioactive compounds, particularly polyphenols. These compounds significantly contribute to the polyphenol intake in the diet and have been shown to have beneficial effects on consumer health. The objective of this research was to conduct a comparative analysis of the polyphenolic composition of coffee beans and infusions obtained from coffee beans sourced from both organic and conventional farming practices while taking into consideration variations in roast intensity and geographical origin. The lyophilized coffee grounds and infusions derived from these grounds were also subjected to analysis. The antioxidant activity was measured by using the radical ABTS, and the quantitative and qualitative analysis of polyphenolic compounds was conducted by HPLC. The conventional coffee samples were richer in chlorogenic acid, catechin, and caffeic acid. However, the coffee beans from organic farming contained more gallic acid, epigallocatechin gallate, and quercetin than those grown conventionally. We did not observe significant differences among the coffee plant production sites in Ethiopia, Sumatra, and Peru, but Peru had the poorest amount of polyphenols when compared to Ethiopia and Sumatra. Coffee infusions prepared from organic coffee beans were characterized by a significantly high sum of identified polyphenols. A higher content of caffeine was observed in the organic coffee bean samples than in the conventional coffee bean samples. Conventional coffee beans were characterized by stronger antioxidant activity than organic beans. Coffees from different parts of the world were characterized by different profiles of polyphenol compounds. Moreover, the coffee beans from Ethiopia were characterized by the highest caffeine content. However, among the different geographical areas of coffee beans, the highest antioxidant activity was detected in the coffee beans from Sumatra. Coffee grounds also have the potential to be used as compounds for the cultivation of horticultural plants, and they can be used as a source of numerous health-promoting compounds in the food and cosmetics industries.

## 1. Introduction

Coffee is a widely consumed beverage worldwide and holds significant value both in international trade and domestic supply due to its substantial quantity and value [1]. A high level of consumption results in a significant volume of waste post-brewing [1,2,3,4,5]. The distribution of coffee demands a considerable degree of processing expertise and leads to the generation of substantial volumes of processing byproducts, exemplified by spent coffee grounds. Global consumption patterns indicate a continuous annual growth rate of 2.2% in coffee consumption worldwide [6]. Geographically, coffee production is confined to the “coffee bean belt,” which is situated between the tropics of Capricorn and Cancer. The three largest coffee-producing countries are Brazil, Vietnam, and Colombia. While coffee cultivation serves as a source of income and employment for millions of households, over 90% of coffee is exported in the form of green beans, and the added value in the coffee industry primarily lies in the importing countries. The period of developing the coffee market and industry can be divided into “three waves of coffee consumption” [7]. In the 1960s, the first wave of coffee consumption emerged, which was marked by exponential growth in consumption, wide availability, and mass market appeal. The second wave of coffee consumption began in the 1990s with the rise of coffeehouse chains, particularly Starbucks, which introduced specialty coffee to cater to consumers’ growing interest in the quality of coffee. This shift transformed coffee from a commodity to a luxury product [8]. The third wave of coffee originated from small roasters who emphasized specific regions and introduced new brewing techniques. Currently, coffee is regarded as a high-quality artisanal product that is often comparable to wine. The act of drinking coffee has transcended its function as a mere beverage and has become associated with pleasure, lifestyle, experience, and social status. This transformation in consumer behavior has been made possible by three approaches that currently define coffee as a consumer product: pleasure, health, and sustainability. Coffee is a beverage that contains a high concentration of bioactive compounds, particularly polyphenols, such as phenolic acids. The most abundant phenolic acids found in coffee beans are chlorogenic acid, which is present in green beans, and caffeic acid, which is formed during the roasting process. Other phenolic acids that can be found in coffee beans include ferulic acid and p-coumaric acid. These compounds significantly contribute to the total polyphenol intake in the diet and have been shown to have beneficial effects on consumer health [9]. Organic farming methods provide solutions and options for the health and environmental issues associated with conventional production technologies and practices. They also incorporate various alternative concepts, including alternative retail and distribution networks, and align with the values of the wholefood movement [10,11,12]. One of the primary benefits of organic farming is the improvement of safety and health, achieved by minimizing pesticide contamination and residues. Additionally, organic farming helps to address concerns related to antibiotic resistance and occupational health hazards, as farm workers are less exposed to chemical pesticides [13,14]. Another advantage of organic farming practices is their positive impact on the environment. Organic farming methods help to conserve the environment by reducing soil erosion and decreasing fertility, which enhances soil fertility. Additionally, they reduce the risk of pesticide pollution in water bodies, provide better ecosystem services than conventional agriculture, and utilize less energy. Studies have shown that organic coffee contains higher levels of polyphenols and other bioactive compounds than conventionally grown coffee. These compounds are believed to provide potential health benefits and have antioxidant and anti-inflammatory properties [9]. In recent decades, numerous in vitro and in vivo studies have highlighted the positive impact of a diet rich in polyphenols or polyphenol-rich products on human health. The consumption of polyphenols has been linked to a lower risk of chronic diseases. The scientific literature contains several studies that explore the physicochemical properties of polyphenols and their mechanisms of action in the prevention of chronic diseases. This review provides a systematic summary of the classification, sources, and components of dietary polyphenols, as well as their efficacy in managing chronic conditions such as diabetes, obesity, hypertension, and hyperuricemia [15]. The impact of caffeine on creative problem-solving aligns with previous research on caffeine and cognition. Studies have consistently demonstrated that caffeine consumption leads to heightened concentration and improved attentional focus, which can serve as mechanisms for improving convergent problem-solving tasks that involve correct solutions. Additionally, caffeine’s impact on higher-order cognitive processes, such as enhanced response inhibition and better performance on selective visual attention tasks, indicates that increased prefrontal activity may support convergent problem-solving. Interestingly, caffeine did not show any differential effects on insight versus analytical solutions, suggesting that it enhances convergent problem-solving overall, regardless of the approach taken [16,17,18]. Caffeine is a commonly used supplement on competition days, as it has been shown to enhance athletic performance by improving cognitive and psychological factors [19].

The objective of this research endeavor was to conduct a comparative analysis of the polyphenolic compositions of coffee beans and infusions obtained from coffee beans sourced from both organic and conventional farming practices while taking into consideration geographical origin. The freeze-dried coffee grounds and infusions derived from these grounds were also subjected to analysis. Coffee grounds are garnering increasing attention from researchers. This study aimed at facilitating enhanced utilization of this raw material and post-production waste, aligning with contemporary global imperatives concerning sustainable development.

## 2. Results

### 2.1. Antioxidant Capacity

The following charts present the results of antioxidant strength measurements for various coffee bean varieties, coffee grounds, and infusions from Ethiopia, Peru, and Sumatra (Figure 1). Coffee samples from both organic and conventional cultivations were collected. In this study, conventional coffee beans were characterized by greater antioxidant power than organic beans (Figure 1). Moreover, among the different geographical areas of coffee beans, the highest antioxidant activity was detected in the coffee beans from Sumatra. In addition, in this study, the antioxidant activity of the conventional coffee infusion was greater than that of the organic coffee infusion (Figure 1). Therefore, coffee brewed in Ethiopia was characterized by the highest antioxidant activity. However, no significant differences in antioxidant activity were found between the organic and conventional coffee grounds. It was observed that infusions of coffee beans from Sumatra had the lowest antioxidant activity. However, after brewing, this coffee had the strongest antioxidant activity.

### 2.2. Caffeine Content 

A higher content of caffeine was observed in the organic coffee bean samples than in the conventional coffee bean samples (Table 1). Moreover, the coffee beans from Ethiopia were characterized by the highest caffeine content. The differences were not statistically significant. On the other hand, the infusions of conventional coffee beans contained more caffeine (Table 2). However, the highest content of caffeine was found after the infusion of coffee beans from Peru. Additionally, we analyzed coffee grounds after the infusion of coffee beans was prepared. In this research, a higher caffeine content was detected in the organic coffee ground samples than in the conventional coffee ground samples, and the coffee grounds from Ethiopia had the highest caffeine content (Table 3).

### 2.3. Polyphenol Content

The content of polyphenol compounds in the coffee beans was also measured. We observed that the conventional coffee samples were characterized by a higher sum of polyphenols as well as individual polyphenol compounds, as follows: chlorogenic acid, catechin, and caffeic acid. However, the coffee beans from organic farming contained more gallic acid, epigallocatechin gallate, and quercetin than those grown conventionally (Table 1). We did not observe significant differences among the coffee plant production sites in Ethiopia, Sumatra, and Peru (Table 1), but Peru had the poorest amount of polyphenols when compared to Ethiopia and Sumatra. Coffee infusions prepared from organic coffee beans were characterized by a significantly high sum of identified polyphenols (Table 2). Coffee grounds are a valuable source of polyphenolic compounds. In this study, the sum of identified polyphenol concentrations did not differ between the organic and conventional coffee or the waste from the origin of the coffee (Table 3). On the other hand, some polyphenols remained in the coffee grounds. In the present study, we did not observe any significant differences in the polyphenol contents between organic and conventional coffee grounds or between coffee of different origins.

### 2.4. PCA 

Based on the PCA results, the overall degree of variability explained by F1 and F2 was 99.15% for the examined coffee processing approaches, coffee origins, and production methods as well as chemical analysis of the coffee products (beans, brew, and grounds) and polyphenol and caffeine contents. This result was confirmed by a strong link between the chemical compositions of the created groups for coffee products and measured features, such as the antioxidant activity, sum of identified polyphenols, caffeine, and antioxidant activity. All examined coffee products were divided into three different groups. The first group comprised qualified coffee beans, both organic and conventional. These coffee products showed a strong link with some chemical compounds, such as caffeine and the caffeic acid content. Moreover, we observed a strong relation with the quercetin content. The second group of relations we observed was between the antioxidant activity, sum of polyphenols, and gallic, chlorogenic, and caffeic acids, mostly only for coffee brews. The third separated group was created for organic and conventional coffee grounds from the three places of origin. Note that the three coffee products are located in three different areas of the chart (next to, to the right of, and to the left of), as previously described. Additionally, it is worth emphasizing that only the coffee infusion from Peru was separated from the other tested coffee brews. The group of coffee grounds did not show any interactions at a significant level (Figure 2).

## 3. Discussion

### 3.1. Polyphenols in Coffee

Coffee is one of the most popular products and infusions worldwide. In our diet, coffee can be treated as a good source of polyphenol compounds [9,20,21,22]. According to a previous study, different forms of chlorogenic acid were identified in coffee beans, but mostly in green coffee beans [23]. In the present study, not green but roasted coffee beans, infusions, and coffee grounds were used as experimental objects. After roasting, different polyphenol compounds appear in coffee products [1]. Organic coffee beans contained fewer polyphenols, but the difference was not statistically significant (Table 1). The obtained results were the opposite of those presented previously by others [1,9]. One of the main rules of organic agriculture is synthetic pesticides’ exclusion. Another name for the polyphenols are “natural pesticides”, produced by plants to protect them against pest attack [24]. In such situations, organic plants produce more polyphenols. In our experiment, a lower level of polyphenols was obtained in organic coffee. Such a situation in organic coffee beans could be an effect of other stress conditions during coffee cultivation. According to the rules on coffee plantations, production bushes should be cultivated in spots where they are shadowed by native plants. This minimizes environmental light stress, leading to lower polyphenol concentrations. A similar tea plant reaction was observed, with lower concentrations of polyphenols in shaded bushes on tea plantations than in plants cultivated with full sunlight [25,26]. However, the coffee products for daily consumption are beans after roasting. This type of coffee processing can change the ratio and the content of total polyphenols in the final product. Another factor influencing the polyphenol composition in coffee beans is coffee origin. In our experiment, three different coffee origins were compared. We did not observe significant differences among the coffee plant production sites in Ethiopia, Sumatra, and Peru (Table 1). However, Peruvian coffee was the poorest in polyphenols compared to Ethiopian and Sumatran coffee beans. In another experiment with Peruvian and Ethiopian coffee, it was shown that coffee beans from Peru contained significantly fewer polyphenols than Ethiopian coffee beans [1,27]. This situation could be an effect of geographical localization of coffee plantations. Ethiopian coffee comes from the Sidamo region. Coffee bushes are grown at altitudes of 1500 to 2200 m above sea level. This region receives abundant rainfall, optimal temperatures, and fertile soil. Peruvian coffee, on the other hand, came from the Ayacucho region with lower rainfall. The plantation was located at an altitude of 1500 to 2000 m above sea level, and the soil fertility was worse compared to the plantation in Ethiopia. In our experiment, organic coffee beans from Peru were characterized by a lower polyphenol concentration compared to conventional ones (Table 1). There are no experiments we can refer to for similar comparisons with coffee beans. A similar study but only with organic cocoa beans showed that a product organically farmed in Peru contained fewer polyphenols (2778 mg/100 g) compared to Columbian beans (3776 mg/100 g). The examined samples from Peru and Columbia were from organic production systems [28]. In the case of roasted coffee beans’ quality and the polyphenol content in the material, the time and temperature of roast processing play a crucial role. A long-term high-temperature roasting process decreases the level of polyphenols in coffee beans (light stage 54.29 mg/g GAE, medium stage 52.14 mg/g GAE, dark stage 42.86 mg/g GAE). Similar findings were presented in another experiment. Two arabica coffees varieties, Burbon and Caturra, had decreasing total polyphenols over time when a deep roasting process was used (light stage 75.53 mg/g GAE, medium stage 62.01 mg/g GAE, and dark stage 50.45 mg/g GAE for Burbon var. and light stage 63.97 mg/g GAE, medium stage 53.17 mg/g GAE, and dark stage 45.26 mg/g GAE for Caturra var.) [29,30].

Coffee infusions prepared from organic coffee beans were characterized by a significantly high concentration of polyphenols (Table 2). The obtained results are supported by others. Organic coffee brews contain significantly more polyphenols than conventional brews, at 104.2 mg/100 mL of liquid coffee and 81.8 mg/100 mL of liquid coffee, respectively [31]. A similar situation was observed with cocoa beans and their infusion. After brewing, organic beverages are richer in total polyphenols than are conventional beverages [32]. The time of infusion is one of the most important factors in the duration of beverage preparation. After hot water penetration, the polyphenols were washed for infusion. It seems, in the case of polyphenols, that short-term coffee brewing is better than long-term brewing [1,23,33]. This balance in the polyphenol concentration at the time of coffee brewing was supported by other studies, such as that on the brewing time of cascara tea [34].

After coffee brewing, waste is left as coffee grounds. Coffee grounds are a valuable source of bioactive compounds with high anti-inflammatory and antioxidant potential. In our experiment, organic and conventional coffee, as well as coffee origin waste, did not differ in the sum of polyphenols (Table 3). However, some polyphenols remained in the coffee grounds. In the present experiment, we did not observe any significant differences in the sum of polyphenols between the organic and conventional coffee grounds (Table 3), as well as different coffee origins. The content of polyphenols in coffee grounds depends on many factors. One of them is coffee bean roasting. Compared with medium-roasted coffee grounds, blondie-roasted coffee grounds contain more polyphenols [35]. Coffee grounds are still a good source of polyphenol compounds. Standard ethanoic acid extraction resulted in more than 17 mg/g polyphenols in the examined coffee grounds [36]. On the basis of our observations, we conclude that in situations allowing for food waste re-use, coffee grounds could make good products for horticulture practices. In a previous study, lettuce plants cultivated with fertilizer had greater biomass but lower micronutrient contents. The results offered a possible solution for the use of coffee grounds as organic amendments by vermicomposting and biocharization to eliminate the toxicity of some metals in soil [1]. In another experiment, lettuce and cucumber cultivated with organic coffee ground amendments showed better quality parameters compared to other fertilizer combinations [37,38,39]. Moreover, spent coffee grounds (SCGs) represent a significant food waste residue, which could be used for weed control in organic agriculture [40]. An experiment with coffee grounds showed that water extract did not inhibit the growth of phytopathogenic fungi or foodborne pathogenic bacteria. However, the extract presented allelopathic activity by inhibiting plant seed germination and reducing seed germination parameters and the germination speed index. Thus, the results indicate that the aqueous extract of coffee grounds has the potential to be used as a natural organic crop herbicide, especially in organic agriculture [41].

### 3.2. Caffeine in Coffee

One of the most important factors in coffee use is the caffeine content. The quantity of coffee beverages consumed by consumers around the world is increasing because of their body and brain stimulation by caffeine [42,43]. In the present study, no significant differences in caffeine content were detected between organic and conventional coffee beans (Table 1). The obtained results are the opposite of others. The organic coffee beans contained lower levels of caffeine compared to conventional methods [22,44]. A similar situation was observed for the geographical area of the coffee beans. No effect of country of origin on the caffeine concentration was observed (Table 1). Previously, coffee beans from Africa were the richest in caffeine when compared to Asian and American beans [45]. In the present study, Ethiopian coffee beans contained the highest levels of caffeine when compared to Sumatran and Peruvian beans (Table 1). The present findings are the opposite of those of other experiments. Coffee beans from Ethiopia contained less caffeine (824.2 mg/100 g beans) than did those from Peru (935.1 mg/100 g beans) [46,47]. According to information from plantations, the same species and variety of coffee were used in the experiment, so we believe that this effect is the result of geographical differences in the cultivation of coffee beans.

Coffee beverages prepared from organic beans were characterized by a lower level of caffeine than conventional beverages (Table 2). This is because coffee beans from organic production are also pure in their caffeine contents. The obtained results were supported by other experiments [48,49,50]. Caffeine is an alkaloid. According to the C/N theory, conventional products produce more N-containing metabolites, including caffeine [51]. In the present experiment, coffee brewing from Peru was characterized by a higher caffeine content. Similar findings were presented by others. Coffee beverages prepared from Peruvian beans were characterized by the highest content of caffeine when compared to that of Ethiopian beverages (50.6 mg/100 mL and 16.7 mg/100 mL, respectively) [51]. The different coffee beverages were grouped together. Compared with Brazilian coffee, the Peruvian brew was characterized as having a high caffeine content (1565.3 mg/100 g and 1454.3 mg/100 g, respectively) [1]. The roasting time negatively affected the caffeine level in a coffee brew. The light-roasted coffee contained 1.095 g/L of caffeine, medium 1.056 g/L, and dark only 1.036 g/L of caffeine. Similar findings were presented by Tfouni et al. (2013) [52,53].

In coffee grounds, a low amount of caffeine is found. It is possible to use pure coffee grounds in agriculture as soil-life stimulators and soil amendments. The level of toxicity of caffeine must be considered. In some experiments, a substantial reduction in caffeine content in the substrate was reported within 50 days of cultivation, without reducing the productivity of the fungus (*Pleurotus ostreatus*). This finding suggested that using coffee grounds without any detoxification pretreatment could offer a feasible alternative for producing mushrooms for human consumption [52]. Caffeine purified from coffee grounds could be used as a pure functional food, for beverage production, as well as in cosmetics [54,55,56].

### 3.3. Individual Phenolics in Coffee

Chlorogenic acid and its derivative are among the most important and characteristic phenolic compounds in coffee beans. Green coffee, before roasting, contains between 13.07 and 22.14 g/100 g of raw beans of total chlorogenic acid [57]. Another study reported lower values of 4.00–4.24 g/100 g raw green coffee beans [58]. The roasting process decreased the concentration of CGAs in the coffee beans. According to the light-roasting data, 60% of the sample was lost from chlorogenic acid, medium 68%, dark 87%, and very dark approximately 98% [59]. Similar findings were presented in other experiments. In green coffee beans, 5.20 g/100 g d.w. of chlorogenic acid was reported. However, in medium-roasted beans, only 1.08 g/100 g d.w. and in the dark 0.59 g/100 g d.w. of chlorogenic acid were reported [60]. According to the present data, conventional coffee contained significantly more chlorogenic acid compared to organic compounds (Table 1). These results are similar to those for other organic and conventional coffee beans [9]. In different experiments, the organic coffee beans were richer in chlorogenic acid compared to conventional ones (54.09 mg/100 g d.w. and 36.94 mg/100 g d.w.) [50]. It seems that the coffee production region influenced the chlorogenic acid content of the roasted coffee beans. Ethiopian coffee beans were characterized as the product with the highest chlorogenic acid concentration (Table 1). Other studies supported our observations. Three coffee origins were included in the group. Ethiopian coffee contains 359.3 mg/100 g d.w., Columbian coffee contains 335.1 mg/100 g d.w., and Sumatra coffee contains only 290.0 mg/100 g d.w. of chlorogenic acid [61]. Another experiment showed that coffee from Peru was the purest in chlorogenic acid (2.78 mg/g) when compared to Sumatra (4.14 mg/g) coffee beans [62]

After coffee beverage preparation, the situation was changed. The organic coffee brewed contained significantly more chlorogenic acid than did conventional brewed coffee brewed. Similar results were reported in other experiments with different coffee brews [50]. The coffee beverages used were similar to the coffee bean infusion prepared from the Ethiopian product and were characterized by the highest chlorogenic acid concentration (Table 2). Similar results were reported in other experiments. There were three different coffee brew origins. Ethiopian coffee (150.4 mg/100 mL) was used, Costa Rican (119.04 mg/100 mL) and Brazilian (118.22 mg/100 mL of chlorogenic acid) [61]. As was reported by Muzykiewicz-Szymańska et al. (2021), the coffee brew from Peru was characterized by the lowest concentration of chlorogenic acid (5.63 mg/mL) compared to the other experimental coffee brews from Columbia (5.78 mg/mL) and India (7.23 mg/mL) [63]

We did not observe an effect of coffee system production or plant origin on the chlorogenic acid content in the coffee grounds. This compound belongs to the hydroxycinaminic phenolic group. Chlorogenic acid is a strong antioxidant and has anti-inflammatory and anti-obesity properties [64]. The second phenolic acid characteristic for coffee is caffeic acid. This compound is used especially for roasted coffee beans. In our experiment, we observed only the effect of the coffee origin on the caffeic acid content (Table 1). Samples of coffee beans from Ethiopia were characterized by the highest caffeic acid content. Caffeic acid was found in small amounts (0.17 mg/100 g) of Brazilian coffee beans [65]. Coffee brew, in contrast, contained only trace amounts of caffeic acid. The second product of degradation, caffeic acid, was not found in the water of coffee extracts. Caffeic acid is characterized by low thermal stability. Its absence in roasted coffee is not unusual [66]. Coffee brewed from Peru contained 8.80–10.35 mg/100 mL caffeic acid, and that from Brazil contained 2.22–8.84 mg/100 mL caffeic acid [1]. In our experiment, 4.93 mg/100 mL of Peruvian coffee was brewed. However, for Ethiopian coffee brewing, it was 10.10 mg/100 mL. Organic coffee grounds are a very pure source of caffeic acid. As we showed in the experiment, after water extraction and coffee brew preparation, the remaining coffee grounds contained 10 times lower concentrations of caffeic acid than did the infusions (Table 3). Freshly brewed coffee grounds contain only trace amounts of caffeic acid. After composting the coffee grounds, chlorogenic acid derivatives were transformed into caffeic acid [67]. In such situations, compost with decomposed coffee grounds can be used as a perfect tool for fungal growth in horticulture, because of the strong activity of caffeic acid, which means it can be used as a natural fungicide. Among the detected phenolic acids, only caffeic acid was significantly induced in plants infected by *Ralstonia solanacearum* relative to healthy tomato plants [68].

Quercetin is a crucial flavonoid that is abundant in coffee beans, brews, and grounds. In coffee beans, only phenolic acids and coumarins detected in their composition showed moderate antioxidant activity in all assays [69]. Quercetin and its derivatives, such as isocoumarins, appear promising tools to fight against inflammatory diseases as well as cancer [70,71]. Compared with conventional beans, organic coffee beans were characterized by a significantly greater quercetin content (Table 1). A similar situation was shown in other experiments with fresh roasted and stored coffee beans [9]. Coffee from Peru was characterized by a significantly greater level of quercetin than Ethiopian and Sumatran coffees. Our results are the opposite of those presented for Peruvian, Ethiopian, and Colombian coffee. Among these three coffee origins, Colombia (1.36 mg/g d.w.) was the richest in quercetin (Table 1), ahead of Ethiopia (1.13 mg/g d.w.) and Peru (1.01 mg/g d.w.) [1]. Coffee brews prepared from organic beans contained significantly more quercetin (Table 2). Similar findings were presented in experiments involving different methods of organic and conventional coffee roasting and brewing [50]. In the group of coffee brew flavonoids, quercetin plays an important role. The whole pool of total flavonoids was always present in quercetin equivalents. A branded coffee brew contained quercetin in the range of 6.14–12.14 mg/40 mL [72]. The coffee grounds left after coffee brewing contained only trace amounts of quercetin. Thermal processes have a great influence on the availability of quercetin from coffee brews, based on their magnitude and duration of exposure. Raw coffee beans were roasted and subsequently extracted with hot water. In such situations, quercetin is labile to heat degradation. One experiment involved boiling and soaking Brazilian beans at 100 °C with or without draining, which induced a loss percentage of almost 90% of the quercetin [73]. We did not observe any differences in quercetin content between organic and conventional coffee grounds, but the coffee waste from Ethiopian coffee beans was the richest in quercetin. The quercetin present in the coffee was extracted when the beverage was prepared. However, spent coffee grounds are still an important source of this compound [74]. Flavonoids, in particular, are found in quercetin, and it has numerous biological properties, including potent antioxidants, anticarcinogenic, anti-allergic, and anti-inflammatory effects, antimicrobial and antitumor properties, as well as beneficial properties related to neuroprotection [75]. It seems that coffee grounds could offer a source of valuable compounds with potential pharmaceutical applications, as well as in the cosmetic and food industries, and spent coffee grounds are an interesting example of waste valorization in the agri-food industry [76].

### 3.4. Antioxidant Activity

The phenolic compound concentration in coffee beans impacts their antioxidant activity. In fact, both phenolic acids and flavonoids support this biological action. In this study, the conventional coffee beans were characterized by greater antioxidant activity compared to organic compounds (Figure 1A). The antioxidant activity of organic coffee depends on the coffee bean roasting. A lower temperature (160 °C) is more efficient than a high temperature (220 °C) [77]. The coffee origin had an effect on the phenolic content as well as the antioxidant power. Among different coffee beans, the highest antioxidant power was observed for Sumatran beans (269.4 µM Trolox/g). Next was Peru (196.8 µM Trolox/g) and then Dominican beans (85.6 µM Trolox/g) [63]. In our study, the antioxidant activity of conventional coffee was greater than that of organic coffee (Figure 1C). The next parameter deciding the antioxidant activity of coffee beans is the roasting process. The roasting process significantly reduces the antioxidant activity of coffee beans. The highest power was observed for light-roasted beans: 77.41%, and it was lower for the medium stage: 66.31%, and the lowest for the dark stage: 58.56%. Similar findings were shown in another experiment. Burbon var. in the light roasting stage has 27.08 µM/L TE, in the medium stage has 21.14 µM/L TE, and in the dark stage has 15.06 µM/L TE. Meanwhile, a second variety showed a similar reaction (in the light roasting stage had 37.40 µM/L TE, in the medium stage had 31.96 µM/L TE, and in the dark stage had 27.34 µM/L TE) [33].

With coffee brews, the obtained results are the opposite of those presented by others The organic coffee infusion had a greater antioxidant effect compared to conventional methods [50]. However, coffee beans from Sumatra had similar phenolic compound contents. Fewer of them were transformed into infusions. Therefore, coffee brewing from Sumatra was characterized by the lowest antioxidant activity (Figure 1C). The three different coffee-brewed Ethiopian infusions were characterized by the highest antioxidant activity (50.6% inhibition), followed by those from Peru (49.3% inhibition) and Sumatra (47.6% inhibition) [78]. As noted previously, coffee grounds contain polyphenols, phenolic acids, and flavonoids. These compounds increase the antioxidant power of widely produced coffee waste. Seven different espresso compounds were characterized by antioxidant activity in the range of 1.5–185.7 IC50 (µg/mL) [79].

## 4. Materials and Methods

### 4.1. Chemicals and Reagents

ABTS (2,20-azino-bis(3-ethylbenzothiazoline-6-sulfonic acid) diammonium salt (Sigma-Aldrich, Warsaw, Poland); acetonitrile (Sigma-Aldrich, Poland); deionized water (Sigma-Aldrich, Poland); ethylacetate (Merck, Warsaw, Poland); methanol (Merck, Warsaw, Poland); ortho-phosphoric acid (Chempur, Warsaw, Poland); phenolic compound standards (purity 99.5–99.9%), including caffeic acid, chlorogenic acid, gallic acid, catechin, epigallocatechin, quercetin, and caffeine (Merck, Poland); and phosphate-buffered saline (Merck, Warsaw, Poland), were used in this study.

### 4.2. Materials

#### 4.2.1. Coffee Beans

The coffees used in this study were purchased from the Polish coffee wholesaler and were selected based on their country of origin (Sumatra, Ethiopia, or Peru) and type of cultivation (organic or conventional). According to the producer declaration, the degree of roasting was the same for both organic and conventional beans. All coffees were 100% Arabica single-origin, variety ‘Typica’. Sumatran coffees originated from the Ache region (both organic and conventional beans), Ethiopian coffees from Sidamo (both organic and conventional beans), and Peruvian coffees from Ayacucho (both organic and conventional beans). The natural coffee cultivation conditions are presented in Appendix A.

#### 4.2.2. Coffee Brewing Method

The brewing accessories used to prepare the coffee infusions were purchased from the Polish store CoffeeDesk (Warsaw, Poland). A Wilfa Svart WSCG-2 (Poznań, Poland) grinder was used for grinding the coffee beans. To achieve the highest extraction level, the coffee was ground very finely. The coffee beans ground in this way were analyzed. Subsequently, the plastic Hario V60-03 dripper (Crakow, Poland) was used in the brewing process, along with dedicated filters of the same brand made from white paper. Each infusion was prepared in the same manner. A 15 g sample was overflowed in 200 mL of water at 100 degrees Celsius. Subsequently, the resulting coffee grounds were lyophilized (freeze-dried). The samples were freeze-dried with a Labcon Co. (Kansas City, MO, USA) freeze drier with a cooling capacity of 2.5 kg of ice per day. The freeze-drying process was carried out using the following parameters: temperature, −40 °C; and pressure, 0.100 mBar. After freeze-drying, the samples were ground in a Mill A-11 laboratory (IKA^®^-Werke GmbH & Co. KG, Staufen, Germany). Each milled sample was placed into a scintillation vial and stored at −80 °C throughout the period of analysis. The powdered samples were used for the chemical analysis described below. Infusions were then prepared from the resulting freeze-dried coffee grounds using the same method.

### 4.3. Analysis of the Antioxidant Potential Using ABTS^•+^

The samples were homogenized using a laboratory grinder. One gram of each sample was mixed with 50 mL of 80% (*v*/*v*) methanol (Chempur, Piekary Śląskie, Poland). The extraction process was performed in an ultrasonic bath at 30 °C for 15 min. After extraction, the samples were centrifuged using an Eppendorf 5804 R centrifuge at 10,000 rpm for 10 min at 0 °C.

Antioxidant activity was assessed using ABTS radicals (Sigma-Aldrich, Poznań, Poland). The ABTS solution was prepared by dissolving ABTS powder (7 mM/L) with potassium persulfate (2.45 mM/L) (Sigma-Aldrich, Poznań, Poland) and allowing the mixture to stand at 20 °C for 24 h. Before use, the ABTS solution was diluted with PBS (Sigma-Aldrich, Poznań, Poland) to achieve an absorbance of 0.7 ± 0.02 at λ = 734 nm.

For the assay, 50 μL of each test sample solution and 150 μL of the ABTS radical solution were added to the wells of a 96-well polystyrene plate. The reaction proceeded for 6 min, and absorbance was measured at 734 nm using a SpectraMax iD3 reader. Results were expressed as mg ascorbic acid equivalents (VCEAC) for 100 g of coffee beans and grounds or 100 mL of prepared coffee brews. Absorbance values were calculated using the standard curve (y = 0.7157 − 391.1242x; R^2^ = 0.996).

The analysis was performed with 5 replicates [50].

### 4.4. Individual Polyphenols and Caffeine Analysis

The quantitative and qualitative analysis of polyphenolic compounds was conducted by the high-performance liquid chromatography (HPLC) method described earlier in detail by Król et al. [9]. One hundred milligrams of ground coffee bean sample were extracted in 5 mL of 80% methanol. For coffee brew preparing, 2 mL of the infusion solution was taken from the prepared brews of ground coffee beans, and 3 mL of 80% methanol was added. After brewing, 100 mg of freeze-dried ground coffee was also extracted in 5 mL of 80% methanol. The samples in plastic tubs were shaken on a Micro-Shaker 326 M (Marki, Poland). Next, all the samples were extracted in an ultrasonic bath (10 min, 30 °C, 5500 Hz). After 10 min of extraction, the coffee samples were transferred to a centrifuge (10 min, 3780× *g*, 5 °C). One milliliter of coffee was transferred to an HPLC vial and used for examination. For analysis purposes, the following HPLC setups were used: two LC-20AD pumps, a CMB-20A system controller, an SIL-20AC autosampler, an ultraviolet–visible SPD-20AV detector, a CTD-20AC oven, and a Phenomenex Fusion-RP 80A column (250 × 4.60 mm), all from Shimadzu (Shimadzu, Tokyo, Japan). The gradient mobile phase contained 10% (phase A) and 55% (phase B) acetonitrile and deionized water. After the mixing of acetonitrile and water in appropriate proportions, orthophosphoric acid (85%) was added, and the pH of the solution was measured simultaneously. After reaching a stable value (3.0), the phases were ready to flow: 1 mL min^−1^, time program: 1.00–22.99 min—phase A 95% and 5% phase B, 23.00–27.99 min—phase A 50% and 50% phase B, 28.00–30.99 min—phase A 80% and 20% phase B, 31.00–42.00 min—phase A 95% and 5% phase B. The wavelengths used for detection were 250 nm for phenolic acids and caffeine (caffeic acid, chlorogenic acid, gallic acid) and 370 nm for flavonoids (catechin, epigallocatechin, quercetin), which were identified based on Fluka and Sigma Aldrich (Warsaw, Poland) external standards with a purity of 99.5%. Examples of chromatograms from polyphenols’ identification are presented in the Appendix A. The results of individual polyphenols were expressed in mg/g of coffee beans and grounds or mg/mL of coffee brews.

### 4.5. Statistical Analysis

The results obtained from chemical measurements were statistically evaluated with Statgraphics Centurion 15.2.11.0 software (StatPoint Technologies, Inc., Warranton, VA, USA). The values presented in the tables and figures are expressed as the mean values for organic and conventional coffee production and their countries of origin (Sumatra, Ethiopia, and Peru). The statistical calculations were based on two-way analysis of variance with the use of Tukey’s test (*p* = 0.05). A lack of statistically significant differences between the examined groups is indicated by similar letters. The standard deviation (SD) is provided with each mean value reported in the tables.

## 5. Conclusions

### 5.1. Findings

Coffee beans are a very good source of polyphenols in our diet. Moderate coffee consumption provides our body with valuable antioxidant compounds. On the basis of the obtained results, organic coffee beans contain fewer individual phenolics compared to conventional ones. Among different country origins, Ethiopian coffee beans were the best in terms of phenolic content. The organic coffee brew was characterized by higher concentrations of chlorogenic acid and quercetin comparing to conventional ones. Additionally, organic coffee grounds were characterized by higher concentrations of catechin and caffeic acid. The experiment showed differences in chemical composition between coffee samples from the two production systems (organic and conventional) and from different countries.

### 5.2. Future Directions for Research

However, further research is necessary on the impacts of other factors affecting the tested products, such as infusions and coffee grounds. In the future, a significantly important experimental direction will be a continuation of the presented research, with the development of a method for the use of health-promoting products obtained from waste coffee grounds, such as coffee ground press oil.

### 5.3. Practical Recommendations for Coffee Producers and Consumers

A practical indication for coffee producers is that the best coffee beans in terms of health benefits are those from Ethiopia. It is worth choosing organic coffees because they have a higher health-promoting value and contribute to promoting consumers’ health. When interpreting the results obtained, it is worth noting that coffee production in Ethiopia takes place on small family farms. Selecting organic coffee from Ethiopia thus supports domestic production and sustainable coffee trade trends.

## Figures and Tables

**Figure 1 molecules-30-01290-f001:**
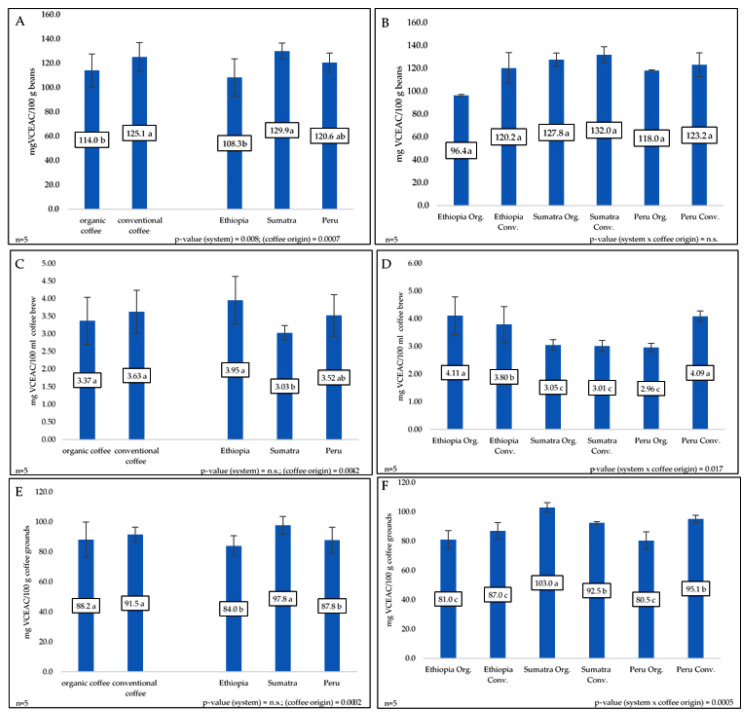
Antioxidant capacity in mg of ascorbic acid equivalents (VCEAC) for 100 g/100 mL of examined coffee beans samples (**A**,**B**), coffee brews (**C**,**D**), and coffee grounds (**E**,**F**). Each bar represents the average value obtained from 5 replications (*n* = 5); different letters mean statistical significant differences at α = 0.05.

**Figure 2 molecules-30-01290-f002:**
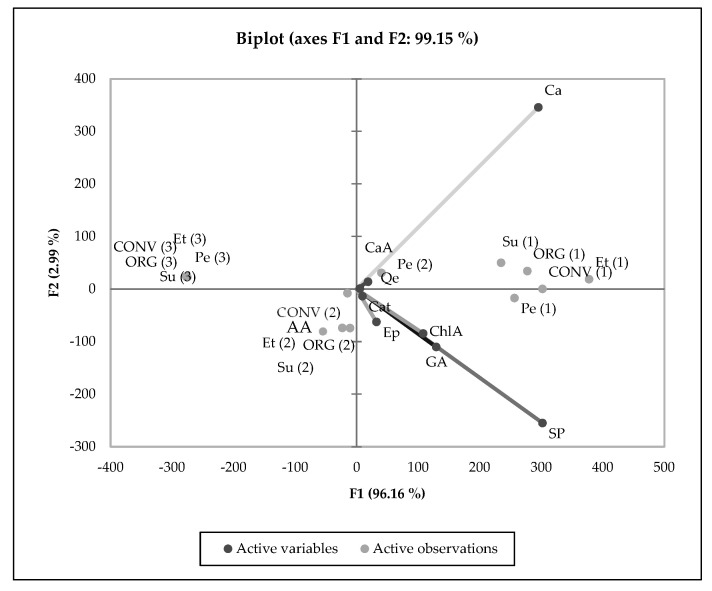
PCA showing the relationship between the chemical compositions and different coffee products, agricultural practices, and coffee origins; sum of polyphenols (SP), gallic acid (GA), caffeine (Ca), chlorogenic acid (ChLA), catechin (Cat), caffeic acid (CaA), epigallocatechin (Ep), quercetin (Qe), antioxidant activity (AA), ORG (organic coffee), CONV (conventional coffee), Pe (Peruvian), Su (Sumatran), Et (Ethiopian) coffees, (1) coffee beans, (2) coffee brews, (3) coffee grounds.

**Table 1 molecules-30-01290-t001:** The contents of caffeine and polyphenols [mg/g] in the examined coffee bean samples.

Compound Name		Coffee Production System				Origin					*p*-Value Production System	*p*-Value Origin
		Organic Coffee		Conventional Coffee		Ethiopia		Sumatra		Peru		
sum of polyphenols		367.29 ± 61.80 a		404.02 ± 103.99 a		439.37 ± 25.42 a		332.32 ± 72.29 a		385.28 ± 108.99 a	NS	NS
gallic acid		173.57 ± 28.10 a		154.70 ± 31.16 a		190.80 ± 6.63 a		153.68 ± 40.29 b		147.92 ± 13.04 b	NS	0.0131
caffeine		368.70 ± 103.56 a		357.75 ± 82.90		420.15 ± 112.41 a		354.25 ± 74.28 a		315.28 ± 58.02 a	NS	NS
chlorogenic acid		108.09 ± 43.23 b		161.81 ± 73.69 a		172.40 ± 14.81 a		85.87 ± 20.03 b		146. 57 ± 94.45 ab	0.0382	0.0263
catechin		11.09 ± 0.58 b		14.14 ± 3.75 a		14.50 ± 14.15 a		10.35 ± 1.27 b		12.99 ± 1.39 ab	0.0082	0.0132
caffeic acid		22.72 ± 1.78 a		24.45 ± 3.06 a		25.35 ± 3.02 a		21.18 ± 1.34 b		24.22 ± 0.85 a	NS	0.0034
epigallocatechin		44.75 ± 18.32 a		42.48 ± 5.58 a		30.62 ± 6.53 c		50.63 ± 11.76 a		43.59 ± 2.18 b	NS	0.0003
quercetin		7.09 ± 2.33 a		6.44 ± 2.69 b		5.69 ± 0.14 a		4.61 ± 0.93 c		9.99 ± 0.49 a	0.02	<0.0001
Interactions												
	sum of polyphenols		gallic acid		caffeine		chlorogenic acid	catechin	caffeic acid	epigallocatechin	quercetin	
Ethiopia org	417.85 ± 4.29 c		193.69 ± 3.61 a		491.43 ± 6.04 a		159.75 ± 5.47 c	10.72 ± 0.21 c	22.72 ± 0.30 bcd	25.28 ± 3.17 d	5.66 ± 0.12 b	
Ethiopia conv	460.88 ± 14.45 b		187.89 ± 8.45 a		348.86 ± 127.70 a		185.04 ± 6.20 b	18.28 ± 0.28 a	27.98 ± 1.37 a	35.95 ± 3.35 c	5.71 ± 0.16 b	
Sumatra org	398.11 ± 7.46 c		190.07 ± 9.01 a		348.62 ± 59.00 a		104.03 ± 2.09 d	10.75 ± 0.28 c	20.71 ± 0.71 e	67.09 ± 1.70 a	5.44 ± 0.19 b	
Sumatra conv	266.53 ± 4.97 d		117.28 ± 1.72 d		359.88 ± 101.08 a		67.71 ± 3.03 e	9.94 ± 1.86 c	21.64 ± 1.82 de	46.17 ± 3.80 b	3.77 ± 0.21 c	
Peru org	285.91 ± 2.88 d		136.93 ± 5.77 c		266.05 ± 19.42 a		60.47 ± 3.35 e	11.79 ± 0.32 c	24.71 ± 0.37 b	41.85 ± 0.45 bc	10.14 ± 0.70 a	
Peru conv	484.64 ± 8.17 a		158.91 ± 5.44 b		364.50 ± 27.72 a		232.66 ± 7.21 a	14.18 ± 0.63 b	23.72 ± 0.97 bc	45.31 ± 1.64 b	9.83 ± 0.17 a	
*p*-value	<0.0001		<0.0001		0.0533		<0.0001	<0.0001	<0.0001	<0.0001	<0.0001	

Mean values, system production (n = 12), coffee origin (n = 6); different letters mean statistically significant differences at α = 0.05. N.S. not significant statistically.

**Table 2 molecules-30-01290-t002:** The contents of caffeine and polyphenols [mg/mL] in the examined coffee brew samples.

Coffee Brew															
Compound Name		Coffee Origin					Geographical Area						*p*-Value Origin		*p*-Value
		Organic Coffee		Conventional Coffee			Ethiopia		Sumatra		Peru				
sum of polyphenols		236.32 ± 26.45 a		196.48 ± 10.99 b			226.85 ± 46.16 a		213.10 ± 18.97 a		209.26 a ± 6.69		0.0008		NS
gallic acid		110.50 ± 13.09 a		104.23 ± 12.64 a			105.55 ± 18.23 a		100.53 ± 6.62 a		116.17 ± 6.10 a		NS		NS
caffeine		99.86 ± 55.27 b		147.87 ± 80.10 a			90.94 ± 21.80 b		67.69 ± 14.29 c		212.95 ± 44.00 a		<0.0001		<0.0001
chlorogenic acid		73.72 ± 35.21 a		45.98 ± 11.09 b			81.35 ± 42.29 a		46.25 ± 12.39 b		51.95 ± 8.94 ab		0.0172		0.0312
catechin		7.53 ± 2.15 a		9.93 ± 4.20 a			10.10 ± 5.50 a		8.29 ± 0.96 a		7.80 ± 2.49 a		NS		NS
caffeic acid		5.15 ± 1.54 b		6.31 ± 0.53 a			6.80 ± 0.45 a		5.46 ± 1.53 b		4.93 ± 0.82 b		0.0121		0.0056
epigallocatechin		36.63 ± 22.31 a		27.93 ± 8.72 a			21.18 ± 8.85 b		50.67 ± 14.78 a		24.99 ± 8.47 b		NS		0.0005
quercetin		2.69 ± 1.21 a		2.12 ± 0.36 b			1.88 ± 0.09 b		1.89 ± 0.12 b		3.44 ± 0.95 a		0.0213		0.0001
Interactions															
	sum of polyphenols		gallic acid		caffeine	chlorogenic acid		catechin		caffeic acid		epigallocatechin		quercetin	
Ethiopia org	268.92 ± 3.99 a		121.68 ± 6.90 a		71.18 ± 3.64 d	119.87 ± 3.59 a		5.09 ± 0.16 d		7.19 ± 0.05 a		13.11 ± 0.81 f		1.96 ± 0.02 cd	
Ethiopia conv	184.77 ± 1.12 d		89.41 ± 1.48 d		110.70 ± 1.73 c	42.81 ± 1.78 cd		15.10 ± 0.63 a		6.40 ± 0.14 b		29.23 ± 0.31 d		1.80 ± 0.00 d	
Sumatra org	229.75 ± 7.96 b		95.06 ± 4.39 cd		55.46 ± 1.81 e	57.24 ± 4.49 b		7.43 ± 0.26 c		4.07 ± 0.21 d		64.11 ± 1.76 a		1.81 ± 0.04 cd	
Sumatra conv	196.43 ± 1.87 cd		105.99 ± 0.75 bc		79.92 ± 7.64 d	35.26 ± 0.95 d		9.14 ± 0.25 b		6.84 ± 0.24 ab		37.22 ± 0.57 b		1.97 ± 0.12 c	
Peru org	210.29 ± 6.54 c		115.06 ± 6.47 ab		172.93 ± 1.60 b	44.03 ± 0.55 c		10.04 ± 0.22 b		4.19 ± 0.18 d		32.66 ± 0.44 c		4.30 ± 0.02 a	
Peru conv	208.23 ± 8.11 c		117.27 ± 6.89 ab		252.97 ± 5.73 a	59.84 ± 3.41 b		5.54 ± 0.55 d		5.67 ± 0.10 c		17.31 ± 1.51 e		2.57 ± 0.03 b	
*p*-value	<0.0001		<0.0001		<0.0001	<0.0001		<0.0001		<0.0001		<0.0001		<0.0001	

Mean values, system production (n = 12), coffee origin (n = 6); different letters mean statistically significant differences at α = 0.05. N.S. not significant statistically.

**Table 3 molecules-30-01290-t003:** The contents of caffeine and polyphenols [mg/g] in the examined coffee ground samples.

Coffee Grounds												
Compound Name		Coffee Origin				Geographical Area					*p*-Value Origin	*p*-Value
		Organic Coffee		Conventional Coffee		Ethiopia		Sumatra	Peru			
sum of polyphenols		7.63 ± 2.72 a		6.65 ± 2.36 a		8.14 ± 0.36 a		6.94 ± 3.73 a	6.35 ± 2.46 a		NS	NS
gallic acid		2.00 ± 0.61 a		2.11 ± 0.99 a		2.73 ± 0.08 a		1.21 ± 0.46 b	2.22 ± 0.73 a		NS	0.0005
caffeine		1.51 ± 0.18 a		1.48 ± 0.23 a		1.73 ± 0.06 a		1.32 ± 0.10 b	1.43 ± 0.11 b		NS	<0.0001
chlorogenic acid		3.15 ± 1.99 a		2.43 ± 1.29 a		2.92 ± 0.10 a		3.15 ± 2.61 a	2.30 ± 1.46 a		NS	NS
catechin		0.94 ± 0.28 a		0.73 ± 0.03 b		0.87 ± 0.16 ab		0.96 ± 0.28 a	0.66 ± 0.08		0.0127	0.0154
caffeic acid		0.77 ± 0.24 a		0.62 ± 0.07 b		0.78 ± 0.09 a		0.78 ± 0.24 a	0.52 ± 0.06 b		0.0201	0.0043
epigallocatechin		0.46 ± 0.17 a		0.46 ± 0.08 a		0.50 ± 0.07 ab		0.52 ± 0.14 a	0.34 ± 0.08 b		NS	0.0266
quercetin		0.32 ± 0.01 a		0.31 ± 0.03 a		0.33 ± 0.01 a		0.31 ± 0.01 b	0.29 ± 0.01 c		NS	0.0002
Interactions												
	sum of polyphenols		gallic acid		caffeine		chlorogenic acid	catechin		caffeic acid	epigallocatechin	quercetin
Ethiopia org	8.34 ± 0.25 b		2.74 ± 0.08 a		1.70 ± 0.04 ab		2.94 ± 0.08 c	1.01 ± 0.4 b		0.86 ± 0.03 b	0.45 ± 0.03 c	0.33 ± 0.01 a
Ethiopia conv	7.93 ± 0.36 b		2.72 ± 0.10 a		1.77 ± 0.07 a		2.90 ± 0.14 c	0.73 ± 0.03 c		0.70 ± 0.04 c	0.55 ± 0.04 b	0.33 ± 0.01 a
Sumatra org	10.34 ± 0.26 a		1.63 ± 0.04 b		1.31 ± 0.11 c		5.54 ± 0.15 a	1.21 ± 0.04 a		0.99 ± 0.02 a	0.65 ± 0.02 a	0.32 ± 0.01 ab
Sumatra conv	3.54 ± 0.27 c		0.79 ± 0.06 c		1.32 ± 0.11 c		0.77 ± 0.08 d	0.71 ± 0.05 c		0.56 ± 0.05 d	0.39 ± 0.03 c	0.31 ± 0.01 ab
Peru org	4.21 ± 0.45 c		1.62 ± 0.47 b		1.51 ± 0.06 bc		0.97 ± 0.03 d	0.59 ± 0.01 d		0.47 ± 0.02 e	0.27 ± 0.01 d	0.30 ± 0.00 ab
Peru conv	8.48 ± 0.24 b		2.83 ± 0.09 a		1.35 ± 0.09 c		3.63 ± 0.12 b	0.74 ± 0.02 c		0.58 ± 0.02 d	0.42 ± 0.02 c	0.28 ± 0.01 b
*p*-value	<0.0001		<0.0001		<0.0001		<0.0001	<0.0001		<0.0001	<0.0001	0.0015

Mean values, system production (n = 12), coffee origin (n = 6); different letters mean statistically significant differences at α = 0.05. N.S. not significant statistically.

## Data Availability

The original contributions presented in this study are included in the article and Appendix A. Further inquiries can be directed to the corresponding author.

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
