# Peer review of "Organic and Conventional Coffee Beans, Infusions, and Grounds as a Rich Sources of Phenolic Compounds in Coffees from Different Origins"

_molecules, 2025, doi:10.3390/molecules30061290_

Round 1

Reviewer 1 Report (Previous Reviewer 3)

Comments and Suggestions for Authors
  1. Figure 1 has the antioxidant activity expressed:”
    mg of ascorbic acid equivalents (VCEAC) for 100 g/100 mL of sample” , can't it be expressed in the same unit of measurement, per 100g for example? (As you specify in materials and methods)  The text below figure 1 contains typos.
  2. In Table 1, 2, 3, polyphenols are expressed in mg/g, does not a single unit of measurement imply all determinations performed for the same sample (e.g. antioxidant activity)?
  3. Table 1 and Table 2 titles, polyphenols not “poliphenols “
  4. In Tables 2 and 3 two terms appear, total polyphenols and sum of polyphenols.
  5. Row 262 “2778 mg/100 g, compare to Columbian beans (3776 mg/100 g ), as well organic [30]” – mg of what?

Row 275 - mg expressed in what?

  1. You have presented in the polyphenol standards compounds such as: kaempferol, kaempferol-3-O-glucoside, quercetin-3-O-glucoside, quercetin-3-O-rutinoside, and salicylic acid. There is no comment regarding them in the results or discussion chapter. Were these compounds identified in the extracts or not?
  2. Specify which polyphenol standards were used in the chromatographic method. Even if the chromatographic method belongs to the authors and has been published previously, its reproduction from the original article in the same form is not allowed. In addition, the bibliographic reference should be the one in which the method is described.

Author Response

Reply to Reviewer no. 1

Thank you very much for the review and for your positive recommendation to publish our manuscript in the Molecules journal. Below you can find answers for your comments and suggestions:

Comment 1: “…Figure 1 has the antioxidant activity expressed: mg of ascorbic acid equivalents (VCEAC) for 100 g/100 mL of sample” , can't it be expressed in the same unit of measurement, per 100g for example? (As you specify in materials and methods)  The text below figure 1 contains typos….”

Authors’ response: Dear Reviewer. Figures 1A and 1B as well as 3A and 3B show the results of analysis coffee beans and coffee grounds. Material for analysis was weighted and automatically the results are calculated on the mass of coffee products and units are 100 g. So the results are presented in unit: mg/100 g. In case of Figures 2A and 2B the object of analysis was coffee infusion. The main goal of those experiment was to check how look like coffee brews after extraction. That is why the results are expressed in mg/100 ml of coffee brews. Theoretically is possible to change units in case of coffee brews, but in such situation it was not have a sense. 15 g of milled coffee beans examined in first part of experiment (Figures 1A and 1B) it was coffee used for coffee brew preparation. On the other hand we want to give information to the consumer, who coffee brews has daily use: how many bioactive compounds he/she intake in 100 ml of coffee brew, not 15g coffee beans used for infusions preparing. For better understanding such situation whole sentence (lines: 526-527) were corrected.

Comment 2: :The text below figure 1 contains typos.”

Authors’ response: the word “eolxamined” was corrected into “of examined”

Comment 3: “…In Table 1, 2, 3, polyphenols are expressed in mg/g, does not a single unit of measurement imply all determinations performed for the same sample (e.g. antioxidant activity)?...”

Authors’ response: The units in which the results are presented are adapted to the given situation. In Tables 1, 2 and 3 the results are given in the unit mg/g. If the unit mg/100 g were chosen, the given values ​​(numbers) would be very large. The authors believe that the clarity of the presentation of the results in tables and figures is also of great importance.

Comment 4: “…Table 1 and Table 2 titles, polyphenols not “poliphenols…“

Authors’ response: The word “polipheols” in Table 1 and Table 2 titles was corrected into “polyphenols”

Comment 5: “…In Tables 2 and 3 two terms appear, total polyphenols and sum of polyphenols….”

Authors’ response: Authors want to apologise. In previous review there was explanation, that: “Authors want to clarified, that in Tables 1, 2, and 3 the sum of counted and identified polyphenols were presented. The table headlines were corrected according this explanation”. Not all table headings were corrected in the past. The Authors verified a very carefully all tables. "Total polyphenols" were corrected into "Sum of polyphenols"

Comment 6: “…Row 262 “2778 mg/100 g, compare to Columbian beans (3776 mg/100 g ), as well organic [30]” – mg of what?”

Authors’ response: Authors want to apologise and they corrected wrong sentence in manuscript text (lines: 260-264)

Comment 7:

Authors’ response: Authors want to apologise and they corrected wrong sentence in manuscript text (line 276)

Comment 8: “…You have presented in the polyphenol standards compounds such as: kaempferol, kaempferol-3-O-glucoside, quercetin-3-O-glucoside, quercetin-3-O-rutinoside, and salicylic acid. There is no comment regarding them in the results or discussion chapter. Were these compounds identified in the extracts or not?...”

Authors’ response: Authors want to apologise. It was typical mistake. Of course only standards for phenolic acids (caffeic acid, chlorogenic acid, gallic acid) and caffeine as well as for flavonoids (catechin, epigallocatechin, quercetin) were used in whole experiment. Those compounds were purchase in Merck and Sigma-Aldrich chemical companies, identified and quantified in experimental material, described in tables and on the chromatograms (Supplementary materials). Information about used standards were corrected unified in whole manuscript text (Tables 1,2 and 3, lines: 479-480; 550-552; Figures 1S-3S (Supplementary materials).

Comment 9: “…Specify which polyphenol standards were used in the chromatographic method. Even if the chromatographic method belongs to the authors and has been published previously, its reproduction from the original article in the same form is not allowed. In addition, the bibliographic reference should be the one in which the method is described...”

Authors’ response: Authors want to apologise. One comment up, the all used, described, discussed and presented polyphenol compounds were listed. The appropriated references was added into manuscript text, where full procedure of analytical details connected with coffee analysis was published before. In presented manuscript, the same HPLC modules as well as other small analytical equipment and analytical condition were used. Of course, Authors not wanted to do plagitizm with before described analytical procedure. I hope now, everything issues are clear.

Reviewer 2 Report (Previous Reviewer 1)

Comments and Suggestions for Authors

The authors have thoroughly addressed the comments raised by the previous reviewer, showcasing attention to detail and dedication to enhancing the manuscript. However, there are two minor points that still require the authors attention. 

1. Table 1: Legend “poliphenols” spelling should be changed to "polyphenols.”

Ref: 71: Ramanan, M.; Sinha, S.; Sudarshan, K.; Aidhen, I. S.; Doble, M. Inhibition of the enzymes in the leukotriene and prostaglandin pathways in inflammation by 3-aryl isocoumarins. Eur. J. Med. Chem., 2016, 124, 428–43. Page numbers should be corrected to 428–434

Author Response

Reply to Reviewer no. 2

Thank you very much for the review and for your positive recommendation to publish our manuscript in the Molecules journal. Below you can find answers for your comments and suggestions:

Comment 1: “1. Table 1: Legend “poliphenols” spelling should be changed to "polyphenols.”.”

Authors’ response: Authors want to apologise. The word “poliphenols” was corrected into “polyphenols”

Comment 2: 71: Ramanan, M.; Sinha, S.; Sudarshan, K.; Aidhen, I. S.; Doble, M. Inhibition of the enzymes in the leukotriene and prostaglandin pathways in inflammation by 3-aryl isocoumarins. Eur. J. Med. Chem., 2016, 124, 428–43. Page numbers should be corrected to 428–434.

Authors’ response: Authors corrected wrong page numbers (line 722)

Round 2

Reviewer 1 Report (Previous Reviewer 3)

Comments and Suggestions for Authors

The authors of the manuscript completed the manuscript in accordance with the reviewers' suggestions.

This manuscript is a resubmission of an earlier submission. The following is a list of the peer review reports and author responses from that submission.

Round 1

Reviewer 1 Report

Comments and Suggestions for Authors

The article titled Coffee beans, infusions, and grounds as rich sources of phenolic compounds explores the polyphenolic composition and antioxidant activity of coffee. It examines coffee beans, infusions, and grounds from both organic and conventional farming practices. The study spans various geographical origins, offering insights into the factors influencing coffee quality. This research highlights the intricate interplay between cultivation methods, origin, and the health benefits of coffee. The authors have conducted a thorough analysis using various analytical techniques to examine the content of total polyphenols, individual phenolic compounds, caffeine, and antioxidant activity in coffee samples from Ethiopia, Sumatra, and Peru. The results presented are in agreement with the findings. The authors have highlighted the potential of coffee grounds as a valuable source of bioactive compounds, suggesting possible applications in horticulture, food, and cosmetic industries. This aligns well with current trends in sustainable development and waste valorization. The geographical comparison further enriches the study, offering insights into how terroir affects coffee composition. The authors have discussed the implications of their findings, particularly in relation to the potential health benefits of coffee consumption and the valorization of coffee waste. However, there are some areas where the study could be improved. Statistical analysis, while adequate, could benefit from more sophisticated techniques to fully explore the complex interactions between farming practices, geographical origin, and coffee processing methods. The authors could also consider expanding on the limitations of their study and suggesting future research directions. Additionally, while the discussion of individual phenolic compounds is thorough, a more integrated analysis of how these compounds interact and contribute to overall coffee quality and potential health benefits would enhance the study's impact. The conclusion section could be strengthened by more explicitly stating the study's main findings and their broader implications for the coffee industry and consumers. The reviewer has the following comments that authors need to address.

1.     The reference numbers cited in the text do not appear, likely due to technical issues. This makes it challenging to identify and align the appropriate references with the corresponding text. The authors should review and address this to ensure clarity.

2.     The authors should include a brief discussion on the potential impact of coffee processing methods, such as roasting time and temperature, on the observed differences in phenolic content and antioxidant activity

3.     As quercetin is a crucial flavonoid abundant in coffee beans, its isomers such as isocoumarins, are well-known for their anti-inflammatory, antioxidant, and neuroprotective properties. The authors are encouraged to include these findings to further emphasize the significance of the phenolic class of compounds in the coffee industry.

https://www.sciencedirect.com/science/article/pii/S0223523416307243

https://www.sciencedirect.com/science/article/pii/S0960894X18310047

4.     It is recommended that the authors include a section on the practical implications of their findings for coffee producers, processors, and consumers. Additionally, providing context on how these findings align with current trends in sustainable coffee production and consumption would enhance the relevance and impact of the study.

5.      A brief discussion on the environmental impacts of organic versus conventional coffee farming practices and expand on the potential applications of coffee grounds in various industries, supported by specific examples or case studies would enhance the impact of the current study.

6.     The manuscript contains minor spelling mistakes. For example, poliphenols. The authors are encouraged to carefully review the entire text and address these errors.

Author Response

Reply to Reviewer no. 1

Thank you very much for the review and for your positive recommendation to publish our manuscript in the Molecules journal. Below you can find answers for your comments and suggestions:

Comment 1: “Statistical analysis, while adequate, could benefit from more sophisticated techniques to fully explore the complex interactions between farming practices, geographical origin, and coffee processing methods..”

Authors’ response: According to Reviewer suggestion, much more advanced PCA analysis have been made, to show the relation between coffee method production, the country origin as well as the coffee processing method. The results of PCA analysis and their description are added to manuscript text and presented on the Figure 2

Comment 2: “The authors could also consider expanding on the limitations of their study and suggesting future research directions. Additionally, while the discussion of individual phenolic compounds is thorough, a more integrated analysis of how these compounds interact and contribute to overall coffee quality and potential health benefits would enhance the study's impact..”

Authors’ response: According to Reviewer suggestion more sophisticated future research direction ideas are added to conclusion section. Moreover in the discussion section more information about individual phenolics health benefits were added into manuscript text (lines: 170-175; 210-215; 237-239.

Comment 3: “The conclusion section could be strengthened by more explicitly stating the study's main findings and their broader implications for the coffee industry and consumers.

Authors’ response: As suggested by the Reviewer, the conclusions of the work have been extended to include practical information for coffee producers and consumers. This information was added into the manuscript text (lines: 370-373);

Comment 4: “The reference numbers cited in the text do not appear, likely due to technical issues. This makes it challenging to identify and align the appropriate references with the corresponding text. The authors should review and address this to ensure clarity.”

Authors’ response: Authors want to apologise. It was technical problem. All bibliographic references now are identifiable in manuscript text.

Comment 5: “The authors should include a brief discussion on the potential impact of coffee processing methods, such as roasting time and temperature, on the observed differences in phenolic content and antioxidant activity.”

Authors’ response:  As suggested by the Reviewer, authors included a new elements for discussion section on the potential impact of coffee processing methods, such as roasting time and temperature, on the observed differences in phenolic content and antioxidant activity (lines 84-93, lines 155-157, lines 261-268)

Comment 6: “As quercetin is a crucial flavonoid abundant in coffee beans, its isomers such as isocoumarins, are well-known for their anti-inflammatory, antioxidant, and neuroprotective properties. The authors are encouraged to include these findings to further emphasize the significance of the phenolic class of compounds in the coffee industry.

https://www.sciencedirect.com/science/article/pii/S0223523416307243

https://www.sciencedirect.com/science/article/pii/S0960894X18310047

Authors’ response: As suggested by the Reviewer, authors add a very important information about  quercetin and their derivatives as isocoumarin into manuscript text (lines 223-227).

Comment 7: “It is recommended that the authors include a section on the practical implications of their findings for coffee producers, processors, and consumers. Additionally, providing context on how these findings align with current trends in sustainable coffee production and consumption would enhance the relevance and impact of the study..”

Authors’ response:  According to Reviewer suggestion section Conclusions were divided into parts contained sub-section “Practical recommendation for coffee producers and consumers” where practical some recommendation and sustainable coffee production and consumption is undelined (lines: 400-405).

Comment 8: “A brief discussion on the environmental impacts of organic versus conventional coffee farming practices and expand on the potential applications of coffee grounds in various industries, supported by specific examples or case studies would enhance the impact of the current study..”

Authors’ response: According to Reviewer suggestion a brief discussion on the environmental impacts of organic versus conventional coffee farming practices and expand on the potential applications of coffee grounds in agriculture were added into manuscript text (lines 119-123)

Comment 9: “The manuscript contains minor spelling mistakes. For example, poliphenols. The authors are encouraged to carefully review the entire text and address these errors..”

Authors’ response: Authors want to apologise. All minor spelling mistakes were corrected in manuscript text.

Reviewer 2 Report

Comments and Suggestions for Authors

The method has many questions to answer so that the experiments can be replicated: Why were these three locations chosen? Were environmental conditions considered? Is it known how each sample is grown? In sun or shade? At which altitudes? Were the beans purchased green, roasted, or ground? How do they control variables such as sample collection dates, and infusion concentrations? Drying methods? Etc. It is necessary to consider climatic, environmental, and sample handling variables, which is somewhat difficult since the samples are purchased from a wholesaler.

The discussion is very general and the explanations for the differences are superficial. This may be because many variables that must be considered in the production of secondary metabolites were not controlled from the beginning, due to how the samples were obtained, which makes it difficult to explain issues such as: why the samples from Peru have fewer polyphenols. Do the environmental characteristics of the sites where the samples were obtained play a role in the differences found? Are these results comparable with those found in other taxa under similar conditions? Yes or no, why? Do organic crops in Peru have more or less polyphenols than conventional crops? Why? etc.

Author Response

Reply to Reviewer no. 2

Thank you very much for the review and for your positive recommendation to publish our manuscript in the Molecules journal. Below you can find answers for your comments and suggestions:

Comment 1: “Why were these three locations chosen?”

Authors’ response: Three locations were selected: Peru, Sumatra and Ethiopia, due to their location at a very similar geographical altitude, parallel to the equator, and simultaneously located on three different continents.

Comment 2: “Were environmental conditions considered? Is it known how each sample is grown? In sun or shade? At which altitudes?”

Authors’ response: The data on the environmental conditions in which the coffee was grown was provided by the producer. The table below presents the data in which the coffee was grown.

temperature

air humidity

crop heigh

rainfall

Peru

15–24°C

56%-76%

1200- 2000 m above sea level

0.3-100 mm

Sumatra

15–25°C

66%-78%

700- 1500 m above sea level

5-1000 mm

Ethiopia

20–25°C

54%-67%

1000- 2100 m above sea level

0-100 mm

Comment 3: “Were the beans purchased green, roasted, or ground? How do they control variables such as sample collection dates, and infusion concentrations? Drying methods? Etc. It is necessary to consider climatic, environmental, and sample handling variables, which is somewhat difficult since the samples are purchased from a wholesaler.”

Authors’ response: The beans were purchased green and then roasted by the producer under the same conditions, to a medium degree. Coffee samples used for analysis come from the September harvest. All this data was obtained from the coffee producer.

Comment 4: “The discussion is very general and the explanations for the differences are superficial. This may be because many variables that must be considered in the production of secondary metabolites were not controlled from the beginning, due to how the samples were obtained, which makes it difficult to explain issues such as: why the samples from Peru have fewer polyphenols.”

Authors’ response: Together with numerous comments from all reviewers, the discussion section has been enriched with many new threads related to the presented research topic. At the same time, other reference4s with similar experiment with a lower content of polyphenol compounds in coffee from Peru was added to the text (lines: 277-285).

Comment 5: “Do the environmental characteristics of the sites where the samples were obtained play a role in the differences found? Are these results comparable with those found in other taxa under similar conditions? Yes or no, why? Do organic crops in Peru have more or less polyphenols than conventional crops? Why? etc.”

Authors’ response:  As other studies indicate, not regarding coffee, but cocoa from Peru, the beans were characterized by a lower polyphenol content. This information was added to the text of the manuscript (lines 260-263).

Reviewer 3 Report

Comments and Suggestions for Authors

The manuscript presents a comparative analysis of the experimental results (individual polyphenol content, antioxidant activity and caffeine content) for 6 coffee samples obtained from organic and conventional crops. The study included both the analysis of coffee beans and that of infusions and grounds after roasting and their use, respectively.

1. The manuscript clearly refers to a comparison between organic and conventional coffee, therefore the title selected for the manuscript does not refer to the main idea of ​​the research.

3. Bibliographic references in the manuscript must be identifiable.

4. The introduction could contain more data regarding similar studies published in specialized journals.

5. “The objective of this research endeavour was to conduct a comparative analysis of
the polyphenolic composition of coffee beans and infusions obtained from coffee beans
sourced from both organic and conventional farming practices while taking into consid-
eration variations in roast intensity and geographical origin.” - Data regarding "variations in roast intensity" are missing from the manuscript.

6. Please specify in what the antioxidant activity was expressed, both in the text and in Figure 1:

"mg/100g beans"

" μM Trolox/g";

"The results are expressed as μM vitamin C equivalent per 10 g of product (μM Tetraethylammonium chloride (TEAC)/10 g of prod-

uct)"

7. Figure 1A, specify how the results were obtained, do these represent the average of several determinations?

8. Are polyphenolic compounds the only compounds that have antioxidant activity in coffee and coffee extracts?

9. In tables 1,2,3 the term Total Polyphenols appears, does this mean that all polyphenolic compounds present in the extracts have been identified or does it only represent the sum of identified polyphenols?

10. What were the arguments for choosing the wavelength value of 250 nm at which certain compounds were quantified? A chromatogram with the identification of the compounds as additional material could be useful.

11. There are a few typos in the tables: "poliphenols", "Etiopis".

12. The units of the results should be unitary: mg/g or mg/100g.

13. In the discussion part, it would be advisable to consider similar studies and to make more comparisons for each type of determinations and results.

Author Response

Reply to Reviewer no. 3

Thank you very much for the review and for your positive recommendation to publish our manuscript in the Molecules journal. Below you can find answers for your comments and suggestions:

Comment 1: “The manuscript clearly refers to a comparison between organic and conventional coffee, therefore the title selected for the manuscript does not refer to the main idea of the research.”

Authors’ response: As suggested by the Reviewer, the title has been changed.

Comment 2: “Bibliographic references in the manuscript must be identifiable”

Authors’ response: Authors want to apologise. It was technical problem. All bibliographic references now are identifiable in manuscript text.

Comment 3: “The introduction could contain more data regarding similar studies published in specialized journals.”

Authors’ response: According to Reviewer suggestion more data regarding studies published in specialized journals have been added into Introduction section.

Comment 4: “The objective of this research endeavour was to conduct a comparative analysis of the polyphenolic composition of coffee beans and infusions obtained from coffee beans sourced from both organic and conventional farming practices while taking into consideration variations in roast intensity and geographical origin.” - Data regarding "variations in roast intensity" are missing from the manuscript..”

Authors’ response: The beans were purchased green and then roasted by the producer under the same conditions, to a medium degree. All coffee samples were roasted to the same degree.

Comment 5: “Please specify in what the antioxidant activity was expressed, both in the text and in Figure 1:

"mg/100g beans"

" μM Trolox/g";

"The results are expressed as μM vitamin C equivalent per 10 g of product (μM Tetraethylammonium chloride (TEAC)/10 g of prod-

uct)"”

Authors’ response: There was a mistake, the mistakes were corrected and now the methodology and units are correct.

Comment 6: “Figure 1A, specify how the results were obtained, do these represent the average of several determinations?

Authors’ response: The obtained results represent the average of several determination. The number of repetitions is presented on each Figure.

Comment 7: “Are polyphenolic compounds the only compounds that have antioxidant activity in coffee and coffee extracts?

Authors’ response: Authors want to clarified, that not only polyphenols have antioxidant activity in coffee beans, brew and ground. According to Reviewer no. 1, comment 6, suggestion, some information about isocoumarins and their health role and chemical properties were added into manuscript text (lines:  lines 223-227)

Comment 8: “In tables 1,2,3 the term Total Polyphenols appears, does this mean that all polyphenolic compounds present in the extracts have been identified or does it only represent the sum of identified polyphenols?

Authors’ response: Authors want to clarified, that in Tables 1,2, and 3 the sum of counted and identified polyphenols were presented. The tables headlines were corrected according this explanation.

Comment 9: “What were the arguments for choosing the wavelength value of 250 nm at which certain compounds were quantified? A chromatogram with the identification of the compounds as additional material could be useful.

Authors’ response: Authors want to explain, that the best wavelength for phenolic acids identification is 250 nm and for flavonoids 370 nm. The using method protocol was developed many years ago, adapted to the presented research conditions, next validated and published in many other research experiments.

Comment 10: “There are a few typos in the tables: "poliphenols", "Etiopis".

Authors’ response: All typos have been corrected.

Comment 11: “The units of the results should be unitary: mg/g or mg/100g.

Authors’ response: All units have now been unified.

Comment 12: “In the discussion part, it would be advisable to consider similar studies and to make more comparisons for each type of determinations and results.

Authors’ response: Thanks for your suggestion, which greatly improved the discussion section. This comment was the same as the previous reviewers' comments and, as suggested, the discussion has been expanded to include consider similar studies and to make more comparisons.
